# SnapMoGen: Human Motion Generation from Expressive Texts

**Chuan Guo[1†],  Inwoo Hwang[2],  Jian Wang[1],  Bing Zhou[1†]**

[1]Snap Inc.     [2] Seoul National University

https://snap-research.github.io/SnapMoGen/

## Abstract

Text-to-motion generation has experienced remarkable progress in recent years. However, current approaches remain limited to synthesizing motion from short or general text prompts, primarily due to dataset constraints. This limitation undermines fine-grained controllability and generalization to unseen prompts. In this paper, we introduce SnapMoGen, a new text-motion dataset featuring high-quality motion capture data paired with accurate, *expressive* textual annotations. The dataset comprises **20K** motion clips totaling 44 hours, accompanied by **122K** detailed textual descriptions averaging 48 words per description (vs. 12 words of HumanML3D). Importantly, these motion clips preserve original temporal continuity as they were in long sequences, facilitating research in long-term motion generation and blending. We also improve upon previous generative masked modeling approaches. Our model, MoMask++, transforms motion into **multi-scale** token sequences that better exploit the token capacity, and learns to generate all tokens using a single generative masked transformer. MoMask++ achieves state-of-the-art performance on both HumanML3D and SnapMoGen benchmarks. Additionally, we demonstrate the ability to process casual user prompts by employing an LLM to reformat inputs to align with the expressivity and narration style of SnapMoGen.

## 1   Introduction

Generating human motions from text has garnered increasing attention in recent years and has experienced notable progress. These advances have been made possible by existing large-scale text-motion datasets [10, 19, 30, 28], and a variety of deep generative models such as VAEs [10, 25], diffusion models [34, 38, 5, 39, 23], GPTs [11, 37, 16], and generative masking [9, 27]. Nevertheless, current models encounter critical limitations when processing complex prompts, falling short in achieving fine-grained control and capturing nuanced variations in human movements. A key contributing factor is the restricted expressivity of text descriptions in existing motion-text datasets. Textual annotations in these datasets are typically brief and general (e.g., "*a person jumps up and lands*"), lacking specific execution details. For instance, in HumanML3D [10], motion sequences of approximately 7 seconds are described by texts averaging only 12 words, which is insufficient to capture motion complexity.

The importance of expressive text annotations has been well-established in other text-conditioned visual content synthesis fields [2, 22, 13, 6, 7]. Descriptive prompts notably enhance the accuracy and aesthetic quality of generated images [22, 2] and improve temporal coherence in video generation [1]. These models understand complex visual content compositions by learning from rich text semantics,

---

[†]Project lead: guochuan5513@gmail.com; bzhou@snapchat.com

39th Conference on Neural Information Processing Systems (NeurIPS 2025).

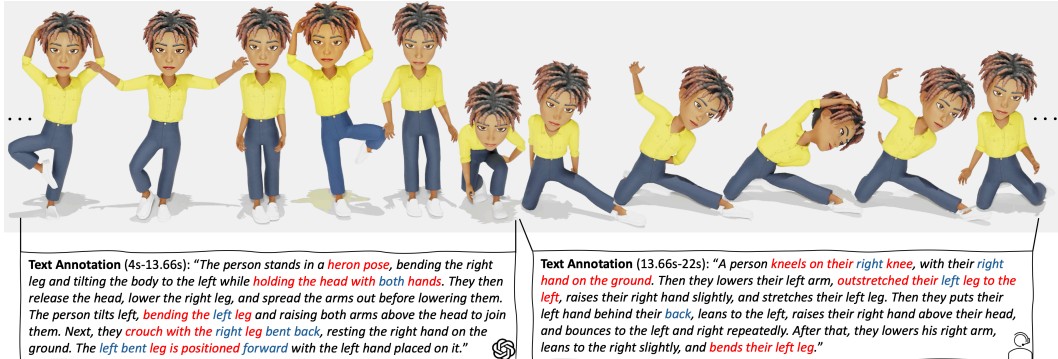

**Text Annotation** (4s-13.66s): *"The person stands in a heron pose, bending the right leg and tilting the body to the left while holding the head with both hands. They then release the head, lower the right leg, and spread the arms out before lowering them. The person tilts left, bending the left leg and raising both arms above the head to join them. Next, they crouch with the right leg bent back, resting the right hand on the ground. The left bent leg is positioned forward with the left hand placed on it."*

**Text Annotation** (13.66s-22s): *"A person kneels on their right knee, with their right hand on the ground. Then they lowers their left arm, outstretched their left leg to the left, raises their right hand slightly, and stretches their left leg. Then they puts their left hand behind their back, leans to the left, raises their right hand above their head, and bounces to the left and right repeatedly. After that, they lowers his right arm, leans to the right slightly, and bends their left leg."*

Figure 1: **Samples from** `SnapMoGen` **dataset.** Our dataset features *temporally continuous* motion segments paired with *highly expressive* text annotations. Each motion clip is accompanied by six distinct textual descriptions. We show an *LLM-augmented* annotation for the first segment, and a *human* annotation for the second.

enabling fine-grained visual content editing [17, 3], adaptation [29, 32, 33], and understanding [4, 36, 18]. More importantly, when dealing with casual user prompts, rich LLM knowledge can be leveraged to enhance prompts with specific details and nuances that models have learned from fine-grained textual training data, effectively improving generalization capability. To foster this research direction in motion synthesis, we introduce `SnapMoGen`, which features high-quality motions captioned with accurate and highly expressive descriptions. `SnapMoGen` is created by segmenting long motion sequences into meaningful 4-12 second clips, each accompanied by six text descriptions—two manually annotated by human experts and four augmented by an LLM that introduces diversity while preserving semantics and temporal consistency. In total, `SnapMoGen` comprises 20K motion clips, amounting to 40 hours of mocap data, accompanied by 122K detailed text descriptions. As shown in Fig. 1, our text annotations contain extremely rich semantic cues of human movements, with an average length of 48 words—three times longer than HumanML3D. Furthermore, our continuous motion segments facilitate research in long-term motion synthesis and motion localization. Tab. 1 presents a statistical comparison between `SnapMoGen` and related motion-text datasets.

To generate motions from expressive texts, we build an improved model upon the previous state-of-the-art approach—MoMask [9]. MoMask applies residual quantization to motion latent features, transforming them into multiple ordered sets of same-length discrete token sequences. Despite achieving pleasing VQ reconstruction through extensive tokens, many of them are not utilized to their full capacity. For instance, tokens following the first quantization layer carry only marginal information. This inefficiency, combined with its layer-specific token vocabulary design, creates inflexibility in subsequent text-to-token generation—necessitating separate models for different token sequences: a primary model for the first sequence and a secondary model for remaining tokens. To overcome these limitations, we adopt a **multi-scale** approach for motion tokenization and generate all motion tokens using a **single** generative masked transformer. In our residual VQ, tokens at each quantization layer focus on a particular temporal scale, following a *coarse-to-fine* gradual progression. Additionally, we share one codebook across all layers to ensure a universal token vocabulary. As shown in Fig. 3, our multi-scale RVQ continually learns meaningful semantics with more layers, outperforming conventional RVQ [9] with $45\%$ less tokens. Then, we simply concatenate all tokens along the temporal dimension and train a generative transformer to produce tokens from text by predicting randomly masked tokens. Our new framework, dubbed **MoMask++**, outperforms MoMask on text-to-motion generation with only a quarter of their token count, as in Tab. 3.

In summary, our key contributions are threefold. First, we introduce `SnapMoGen`, a large-scale dataset comprising 20K temporally continuous motion capture clips described by 122K highly expressive text prompts. We also establish comprehensive benchmarks and evaluation protocols for this new dataset. Second, we advance beyond the existing state-of-the-art approach by proposing MoMask++, which optimizes motion token capacity through multi-scale quantization and models text-conditioned token generation using a single generative masked transformer. Third, we demonstrate effective handling of casual user prompts through LLM-based prompt rewriting, enabled by the descriptive captions in our `SnapMoGen`.

| Datasets | Year | # Clips | Duration | # Texts | Avg. words per text | Avg. length per clip | Mocap? | Continuous? |
|---|---|---|---|---|---|---|---|---|
| KIT-ML [28] | 2016 | 3,911 | 10.3 h | 6,278 | 8 | 9.5s | ✓ | ✗ |
| BABEL† [30] | 2021 | 52,937 | 33.2h | 52,937 | 2 | 2.3s | ✓ | ✓ |
| HumanML3D [10] | 2022 | 14,616 | 28.6h | 44,970 | 12 | 7.1s | ✓ | ✗ |
| Motion-X [19] | 2023 | 81,084 | 144.2h | 81,084 | 9 | 6.4s | ✗ | ✗ |
| SnapMoGen | 2025 | 20,450 | 43.7h | 122,565* | 48 | 7.8s | ✓ | ✓ |

Table 1: **Comparisons with public datasets.** SnapMoGen highlights its accurate and expressive text descriptions, high-quality motion capture data, and continuous motion segmentation. † indicates values calculated only from the publicly available BABEL subset. ∗ denotes a combination of 40,859 manual text annotations and 81,706 LLM-augmented annotations, both with an average text length of 48 words.

## 2 Related Work

**Human Motion-Text Dataset.** KIT Motion-Language Dataset [28] pioneered this domain with 3.9K motions and 6.3K human-annotated descriptions but was limited in scale and text diversity. BABEL [30] introduced temporally precise frame-level labels across 33 hours of motion capture data; however, its annotations primarily consist of short phrases (e.g., 'lift something') for approximately 2-second atomic actions rather than descriptions of extended sequences. HumanML3D [10] expanded the field with 14.6K motions and 44.9K texts by aggregating data from AMASS [21] and HumanAct12 [12]. Despite its size, the text descriptions remain brief and general (e.g., "*a person was pushed but did not fall*"), failing to capture nuanced movement details. Motion-X [19] increased diversity by extracting motions from monocular videos and generating descriptions using video captioning models [36]. However, these motions often contain estimation artifacts such as jitter and foot-sliding, while their descriptions still lack expressivity. Recently, HuMMan-MoGen introduced fine-grained descriptions for specific body parts in motions. In contrast, our SnapMoGen introduces highly expressive text descriptions for holistic 4-12 second motion segments.

**Human Motion Generation.** Recent advances in human motion generation, particularly in text-conditioned synthesis, have significantly improved the realism and text controllability of generated motions. Early methods explored continuous motion representations using generative models such as VAEs [10, 25]. The introduction of diffusion models [34, 38, 5, 39, 23] has significantly advanced the field. By iteratively refining motion through denoising steps, these models generate realistic sequences that align closely with textual prompts. A parallel line of research models motion as sequences of discrete tokens using quantization techniques such as VQ-VAEs [35]. These approaches represent motion as compact, structured token sequences, typically generated autoregressively [20, 15, 11, 37, 16] or through generative masking schemes [9, 27]. To reduce quantization error, MoMask [9] applies multiple quantization layers to iteratively approximate the residuals. Nevertheless, as all quantization is applied at the same (and full) temporal scale, the information captured at each successive layer decreases drastically, leading to an overproduction of tokens with notably uneven information content. This inefficiency also makes text-to-token generation rather inflexible. These limitations directly inspire the multi-scale residual quantization process in our framework.

## 3 SnapMoGen Dataset

SnapMoGen encompasses 43.7 hours of high-quality motion data captured at 30 frames per second. The dataset comprises a total of 4.7M motion frames, featuring a diverse range of actions including daily activities, fitness routines, social interactions, dances, and more. We deliberately incorporate various stylized performances (e.g., princess, elderly person, zombie) to enhance diversity. SnapMoGen captures performances from 10 participants, resulting in 20,450 motion clips ranging from 4 to 12 seconds in length. Each motion clip is accompanied by 6 detailed textual descriptions (2 manually annotated, 4 LLM-augmented), totaling 122,565 textual descriptions with an average length of 48 words. A comparison between SnapMoGen and existing motion-text datasets is presented in Table 1. We further augment the dataset by mirroring motion data [10] throughout our experiments.

**Motion Coverage.** We aim to cover a wide range of actions while ensuring high-quality 3D motion capture. All motions are recorded using Xsens[1] and Rokoko[2] motion capture suits. To determine motion content, we combine two resources: (i) LLM-generated action scenarios covering diverse topics, and (ii) a curated collection of videos and images from the internet featuring content of interest, such as stylized movements. These text instructions and video demonstrations are presented to performers prior to recording sessions as reference material. Performers are then encouraged to execute these or related actions in their own interpretive style. Following data collection, motions with notable artifacts (e.g., jittering, foot sliding) are filtered out to maintain data quality.

**Segmentation.** We deliberately capture long motion sequences containing multiple actions for broader applications. Subsequently, we develop an automated pipeline to segment these sequences into shorter clips of appropriate lengths. The key principle is to prioritize segmentation at motionless moments. Specifically, we first calculate the average positional velocities of hip and end-effector joint at each frame, smoothed by a Gaussian filter. We then detect velocity troughs and normalize their values within each sequence, yielding $\rho_{1:n} \in [0, 1]$, where $n$ indicating the number of troughs. Each trough $i$ is selected as a segmentation point with probability $0.5\rho_i$, which typically results in clips averaging 8 seconds. Hard constraints of minimum (4s) and maximum duration (12s) are enforced during segmentation. Segmentation examples are provided in supplementary files.

**Text Annotation.** Each motion clip is rendered as video using a 3D character for annotation. We collect descriptions from two distinct annotators for each motion clip. The entire annotation process involves 55 professional native English-speaking annotators who are instructed to address the following aspects in their textual descriptions: `action`, `context`, `style`, `moving direction`, `speed`, `trajectory shape`, `body parts`, `spatial relation/location`, `posture` (if applicable), and `timing` (if applicable). All annotations undergo a second-round review to ensure descriptive accuracy. Typographical errors in the collected textual descriptions are corrected using an LLM. To enhance textual diversity, we further employ the LLM to re-describe each manual description twice, maintaining precise action semantics while varying expression. This results in a total of six distinct descriptions per motion clip.

## 4  Method

Our goal is to generate a 3D human pose sequence $\mathbf{m}_{1:N}$ of length $N$ guided by a textual description $c$, where $\mathbf{m}_i \in \mathbb{R}^D$ and $D$ denotes the dimension of pose features.

### 4.1  Preliminary: Motion Tokenization via Residual Quantization

In traditional motion VQ-VAEs [27, 11, 37], a motion encoder $\mathcal{E}(\cdot)$ encodes the motion sequence $\mathbf{m} \in \mathbb{R}^{N \times D}$ to a latent feature sequence $f \in \mathbb{R}^{n \times d}$, which is further mapped to a discrete token sequence $q \in [K]^n$ through vector quantization:

$$f = \mathcal{E}(\mathbf{m}), \qquad q = \mathcal{Q}(f),$$

where $\mathcal{Q}(\cdot)$ denotes a quantizer. The quantizer typically consists of a learnable codebook $\mathcal{C} \in \mathbb{R}^{K \times d}$ of $K$ codes. During quantization, each feature vector $f_i$ is mapped to the code index $q_i$ of its nearest code entry in the codebook:

$$q_i = \left( \texttt{argmin}_{k \in [K]} \| \texttt{lookup}(\mathcal{C}, k) - f_i \|_2 \right) \in [K] \tag{1}$$

where $\texttt{lookup}(\mathcal{C}, k)$ means taking the $k$-th vector in codebook $\mathcal{C}$. The quantized feature vector sequence is finally fed into a decoder $\mathcal{D}$ to reconstruct the input motion:

$$\hat{f} = \texttt{lookup}(\mathcal{C}, q), \qquad \hat{\mathbf{m}} = \mathcal{D}(\hat{f}). \tag{2}$$

To effectively reduce quantization errors, MoMask [9] introduces additional $V$ quantization layers $\mathcal{Q}^{1,..,V}(\cdot)$. Specifically, starting from the initial residual $r^0 = f$, each $\mathcal{Q}^v(\cdot)$ calculates token indices

---

[1] https://www.movella.com/products/motion-capture/xsens-mvn-animate

[2] https://www.rokoko.com/products/smartsuit-pro

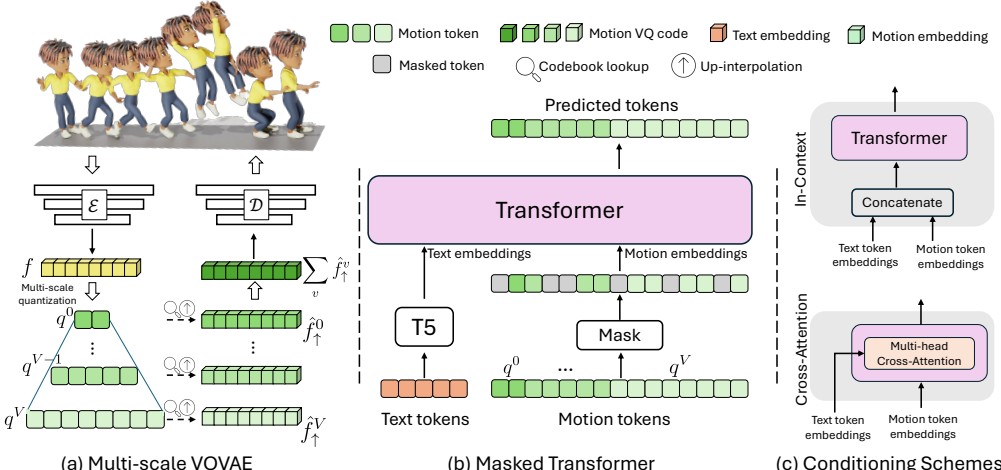

**Figure 2: Approach overview.** (a) A multi-scale VQVAE encodes a motion sequence into $V + 1$ discrete token sequences $(q^0, ..., q^V)$, where each sequence $q^v$ operates at a specific temporal resolution $h^v$. Their corresponding quantized features are upsampled to full resolution $\{\hat{f}^v_\uparrow\}^V_{v=0}$ and summed before being fed into the decoder. (b) A single masked transformer operates on tokens from all scales. Token sequences from a motion are concatenated along the temporal dimension and randomly masked with a variable rate. The transformer is trained to predict the masked tokens conditioned on text and the partially masked sequence. (c) We implement two methods for text conditioning: in-context learning and cross-attention.

$q^v$ and their corresponding codes $\hat{f}^v$ as an approximation of residual $r^v$, and then computes the next residual $r^{v+1}$ as:

$$q^v = \mathcal{Q}^v(r^v), \qquad \hat{f}^v = \texttt{lookup}(\mathcal{C}^v, q^v), \qquad r^{v+1} = r^v - \hat{f}^v \tag{3}$$

Each quantization layer $\mathcal{Q}^v(\cdot)$ contains a *separate* codebook $\mathcal{C}^v \in \mathbb{R}^{K \times d}$. This approach yields $V + 1$ discrete token sequences $[q^v]^V_0 \in [K]^{(V+1) \times n}$ of length $n$ for a motion sequence. The final approximation of the latent sequence $f$ is the sum of all quantized features $\hat{f} = \sum^V_{v=0} \hat{f}^v$.

Overall, this quantized auto-encoder model is trained using a compound loss function that combines motion reconstruction and per-layer latent embedding losses:

$$\mathcal{L}_{rvq} = \texttt{SmoothL1}(\mathbf{m} - \hat{\mathbf{m}}) + \beta \sum^V_{v=0} \|r^v - \texttt{sg}[\hat{f}^v]\|_2, \tag{4}$$

where $\texttt{sg}[\cdot]$ denotes the stop-gradient operation, and $\beta$ a weighting factor for embedding alignment. The codebook entries are updated using exponential moving average [37].

**Discussion.** Although high-fidelity VQ reconstruction can be achieved through an extensive set $((V + 1) \times n)$ of motion tokens, this approach introduces inflexibility in learning the text-to-token mapping, primarily due to two factors: i) information is disproportionately distributed across quantization layers—first-layer tokens typically contain the predominant features, while subsequent layers capture only incremental refinements; and ii) tokens at different layers are indexed by independent codebooks. To address this heterogeneity, MoMask [9] applies an expressive generative masked transformer for the principal first-layer tokens, while modeling all other-layer tokens with a secondary transformer conditioned on first-layer results. This hierarchical approach further diminishes the representational capacity of tokens in non-first layers.

### 4.2 Our Approach: MoMask++

In our approach, tokens at different quantization layers are designed to capture information at specific temporal resolutions, with a common codebook shared across all layers. This design allows us to model the generation of all tokens using a **single** generative masked transformer.

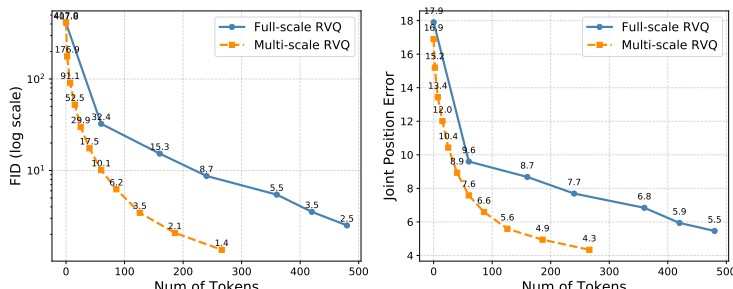

Figure 3: Illustration of token capacity in a pretrained traditional *6-layer, 480-token* full-scale RVQ [9] compared to a *10-layer, 266-token* multi-scale RVQ[3]. Starting from a zero-sequence, we incrementally add one quantized feature sequence for motion decoding and measure the reconstruction performance. The multi-scale VQ learns tokens more efficiently with meaningful semantics at each quantization layer.

### 4.2.1 Multi-scale Motion Quantization

As depicted in Fig. 2 (a), for a motion latent feature sequence $f \in \mathbb{R}^{n \times d}$, our quantizer employs a series of residual quantization operations at progressively increasing temporal resolutions $\{h^v\}_{v=0}^V$, where $h^0 < \cdots < h^V = n$. We denote all quantization operations uniformly as $\mathcal{Q}(\cdot)$ as they share a common codebook $\mathcal{C}$. Quantized features at coarse level $\hat{f}^v \in \mathbb{R}^{h_v \times d}$ are bilinearly interpolated to full resolution $\hat{f}^v_\uparrow = \mathcal{I}(\hat{f}^v, h^V)$, where residuals are calculated and then downsampled to next scale ($h_{v+1}$) to be quantized by the succeeding layer. Mathematically, Eq. (3) is reformulated as:

$$q^v = \mathcal{Q}(\mathcal{I}(r^v, h^v)), \qquad r^{v+1} = r^v - \mathcal{I}(\hat{f}^v, h^V), \qquad r^0 = f, \qquad (5)$$

where $\hat{f}^v = \texttt{lookup}(\mathcal{C}, q^v)$. Then, the final approximation of latent sequence $f$ is the sum of all up-interpolated quantized sequences, which is then fed into decoder $\mathcal{D}$ for motion reconstruction:

$$\hat{f} = \sum_{v=0}^V \mathcal{I}(\hat{f}^v, h^V), \qquad \hat{\mathbf{m}} = \mathcal{D}(\hat{f}).$$

We further add self-attention layers after each res-block in existing motion VQVAE architectures [9, 37] for higher-fidelity motion reconstruction. During training, we introduce additional emphasis on reconstructing essential rotational features $\mathcal{L}_{ess}$ on the top of $\mathcal{L}_{rvq}$ in Eq. (4), weighed by $\lambda_{ess}$. The final learning objective becomes:

$$\mathcal{L}_{ms\_rvq} = \mathcal{L}_{rvq} + \lambda_{ess}\mathcal{L}_{ess} \qquad (6)$$

After all, a motion sequence is represented as $V + 1$ ordered discrete token sequences with a hierarchy of temporal scales $q = (q^0, q^1, ..., q^V)$, where each $q^v$ has a length of $h^v$. Since a shared codebook is utilized across all scales, tokens from each $q^v$ belong to the same vocabulary $[K]$.

As shown in Fig. 3, compared to previous all full-scale residual VQ [9], our multi-scale VQ effectively exploits token capacity and continually learns meaningful semantic features at each quantization layer. It achieves superior reconstruction quality significantly with fewer tokens (266 vs. 480).

### 4.2.2 Learning Text-to-token Mapping

We employ a single bidirectional transformer for token generation from text descriptions. Our framework is illustrated in Figure 2 (b-c). We utilize `T5-base` [31] to extract word-level features from complex textual descriptions $c$. Motion tokens from all scales are concatenated along the temporal axis, yielding an extended token sequence $q$, which is then embedded through an MLP. We investigate two primary architectures for text conditioning: (i) **In-context learning**, where embeddings of motion tokens and text tokens are concatenated and processed uniformly as transformer input, and (ii) **Cross-attention**, which incorporates additional multi-head cross-attention layers that enable motion features to query relevant text features.

---

[3]Token counts are based on encoding a 320-pose sequence. For multi-scale VQ, we use [2, 5, 7, 10, 15, 20, 26, 40, 60, 80] tokens from the 1st to 10th scale.

| Methods | R Precision↑ | | | FID↓ | MM Dist↓ | MModality↑ |
|---|---|---|---|---|---|---|
| | Top 1 | Top 2 | Top 3 | | | |
| Real motions | $0.511^{\pm.003}$ | $0.703^{\pm.003}$ | $0.797^{\pm.002}$ | $0.002^{\pm.000}$ | $2.974^{\pm.008}$ | - |
| TM2T [11] | $0.424^{\pm.003}$ | $0.618^{\pm.003}$ | $0.729^{\pm.002}$ | $1.501^{\pm.017}$ | $3.467^{\pm.011}$ | $\underline{2.424}^{\pm.093}$ |
| T2M [10] | $0.455^{\pm.003}$ | $0.636^{\pm.003}$ | $0.736^{\pm.002}$ | $1.087^{\pm.021}$ | $3.347^{\pm.008}$ | $2.219^{\pm.074}$ |
| MDM [34] | - | - | $0.611^{\pm.007}$ | $0.544^{\pm.044}$ | $5.566^{\pm.027}$ | $\mathbf{2.799}^{\pm.072}$ |
| MLD [5] | $0.481^{\pm.003}$ | $0.673^{\pm.003}$ | $0.772^{\pm.002}$ | $0.473^{\pm.013}$ | $3.196^{\pm.010}$ | $2.413^{\pm.079}$ |
| MotionDiffuse [38] | $0.491^{\pm.001}$ | $0.681^{\pm.001}$ | $0.782^{\pm.001}$ | $0.630^{\pm.001}$ | $3.113^{\pm.001}$ | $1.553^{\pm.042}$ |
| T2M-GPT [37] | $0.492^{\pm.003}$ | $0.679^{\pm.002}$ | $0.775^{\pm.002}$ | $0.141^{\pm.005}$ | $3.121^{\pm.009}$ | $1.831^{\pm.048}$ |
| MMM [27] | $0.515^{\pm.002}$ | $0.708^{\pm.002}$ | $0.804^{\pm.002}$ | $0.089^{\pm.005}$ | $\underline{2.926}^{\pm.007}$ | $1.226^{\pm.040}$ |
| MoMask [9] | $\underline{0.521}^{\pm.002}$ | $\underline{0.713}^{\pm.002}$ | $\underline{0.807}^{\pm.002}$ | $\mathbf{0.045}^{\pm.002}$ | $2.958^{\pm.008}$ | $1.241^{\pm.040}$ |
| MoMask++[in] | $\mathbf{0.528}^{\pm.003}$ | $\mathbf{0.718}^{\pm.003}$ | $\mathbf{0.811}^{\pm.002}$ | $0.072^{\pm.003}$ | $\mathbf{2.912}^{\pm.008}$ | $1.227^{\pm.046}$ |
| MoMask++[cra] | $0.517^{\pm.002}$ | $0.709^{\pm.002}$ | $0.803^{\pm.002}$ | $\underline{0.069}^{\pm.003}$ | $2.948^{\pm.007}$ | $1.192^{\pm.053}$ |

Table 2: **Quantitative evaluation on HumanML3D test set.** $\pm$ indicates a 95% confidence interval. **Bold** indicates the best result, while underscore refers to the second best. "*in*": in-context. "*cra*": cross-attention.

In training, a varying fraction $\gamma(\tau) = \cos(\frac{\pi\tau}{2}) \in [0, 1]$, where $\tau \sim \mathcal{U}(0, 1)$, of sequence elements is uniformly selected, masked out, and replaced with a special [MASK] token. The transformer is trained to predict these masked tokens given text input $c$ and the partially masked token sequence $\dot{q}$, by maximizing the likelihood:

$$\mathcal{L}_{mask} = \sum_{\dot{q}_k=[\texttt{MASK}]} -\log p_\theta\left(q_k|\dot{q}, c\right). \tag{7}$$

We adopt the *replacing and remasking* strategy [9, 8] to enhance contextual reasoning ability. Additionally, the model is trained without text condition $c = \emptyset$ with a probability of 10% to enable classifier-free guidance (CFG).

During inference, a complete sequence of $q$ can be generated in a constant number ($L$) of iterations. This process begins with an empty sequence $[[\texttt{MASK}]]^N$ where all tokens are masked, with $N = \sum_{v=0}^{V} h^v$ denoting the total number of tokens in $q$. At each iteration ($l$), the model predicts categorical token distributions at masked locations, samples tokens, and re-masks the $\lceil \gamma(\frac{l}{L} \cdot N) \rceil$ lowest-confidence tokens. This process repeats until $l$ reaches $L$. We also adopt classifier-free guidance as in [9] with guidance scale $s$. Finally, all generated tokens are decoded and projected back to motion sequence through the VQ-VAE decoder.

# 5 Experiments

Besides `SnapMoGen`, we also conduct experiments on HumanML3D [10], a popular motion-text dataset comprising 14,616 motions with 44,970 textual descriptions.

**Dataset Setup.** We process motions in `SnapMoGen` following procedures established in HumanML3D, including motion mirroring and standardization. To prevent data leakage, we deliberately hold out a test (%10) set and a validation (%5) set where the motion scenarios (e.g., fashion) differ from the training motions. We primarily adopt the feature representation from HumanML3D, consisting of root angular velocity along Y-axis $\dot{r}^a \in \mathbb{R}$, root linear velocity on XZ-plane $\dot{r}^{xz} \in \mathbb{R}^2$, root height $\dot{r}^y \in \mathbb{R}$, 6D local joint rotations $\mathbf{j}^r \in \mathbb{R}^{6j}$, local joint positions $\mathbf{j}^p \in \mathbb{R}^{3j}$, and local joint velocities $\mathbf{j}^v \in \mathbb{R}^{3j}$, where $j$ denotes the number of joints. We empirically find that this comprehensive set of pose features leads to slightly better performance (Tab. 4). Our `SnapMoGen` follows a skeletal topology comprising 24 joints, resulting in 296-dimensional pose features. Unlike HumanML3D, our pose features are directly convertible to standard motion capture file formats (e.g., BVH).

**Evaluation Setup.** We adopt established metrics including FID, R-Precision, MultiModal Distance, and Multimodality following previous works [9, 14, 34]. The evaluator from prior research [10] was exclusively trained to align motion and text embeddings. However, the resulting motion embeddings may be biased toward text alignment while overlooking motion fidelity. Additionally, its redundant motion feature design lacks flexibility for broader evaluation scenarios [23]. Therefore, we adopt

| Methods | R Precision↑ | | | FID↓ | CLIP Score↑ | MModality↑ |
|---|---|---|---|---|---|---|
| | Top 1 | Top 2 | Top 3 | | | |
| Real motions | $0.940^{\pm.001}$ | $0.976^{\pm.001}$ | $0.985^{\pm.001}$ | $0.001^{\pm.000}$ | $0.837^{\pm.000}$ | - |
| MDM [34] | $0.503^{\pm.002}$ | $0.653^{\pm.002}$ | $0.727^{\pm.002}$ | $57.783^{\pm.092}$ | $0.481^{\pm.001}$ | $\mathbf{13.412}^{\pm.231}$ |
| T2M-GPT [37] | $0.618^{\pm.002}$ | $0.773^{\pm.002}$ | $0.812^{\pm.002}$ | $32.629^{\pm.087}$ | $0.573^{\pm.001}$ | $9.172^{\pm.181}$ |
| StableMoFusion [14] | $0.679^{\pm.002}$ | $0.823^{\pm.002}$ | $0.888^{\pm.002}$ | $27.801^{\pm.063}$ | $0.605^{\pm.001}$ | $9.064^{\pm.138}$ |
| MARDM [23] | $0.659^{\pm.002}$ | $0.812^{\pm.002}$ | $0.860^{\pm.002}$ | $26.878^{\pm.131}$ | $0.602^{\pm.001}$ | $\underline{9.812}^{\pm.287}$ |
| MoMask [9] | $0.777^{\pm.002}$ | $0.888^{\pm.002}$ | $0.927^{\pm.002}$ | $17.404^{\pm.051}$ | $0.664^{\pm.001}$ | $8.183^{\pm.184}$ |
| MoMask++[in] | $\mathbf{0.805}^{\pm.002}$ | $\underline{0.904}^{\pm.002}$ | $\mathbf{0.938}^{\pm.001}$ | $\underline{15.56}^{\pm.071}$ | $\underline{0.684}^{\pm.001}$ | $6.556^{\pm.178}$ |
| MoMask++[cra] | $\underline{0.802}^{\pm.001}$ | $\mathbf{0.905}^{\pm.002}$ | $\underline{0.938}^{\pm.001}$ | $\mathbf{15.06}^{\pm.065}$ | $\mathbf{0.685}^{\pm.001}$ | $7.259^{\pm.180}$ |

Table 3: **Quantitative evaluation on SnapMoGen test set.**

the TMR [26] approach for our evaluation model, utilizing only essential 148-dimensional motion features. This method extracts separate latent vectors from motion and text, requiring the motion vector to both align well with corresponding text features and accurately reconstruct the source motion (ensuring fidelity). We use the `T5-base` model to extract word-level text features. For R-Precision calculations, we employ a candidate pool size of 100. We also use the CLIP score [23] to evaluate text-motion alignment, which measures the cosine similarity between text and motion features.

**Comparison Models.** On `SnapMoGen`, We reproduce baseline methods across three mainstream generative paradigms: diffusion models (MDM [34], StableMoFusion [14], and MARDM [23]), autoregressive models (T2M-GPT [37]), and generative masking approaches (MoMask [9]). We utilize their official codebases. Each experiment is repeated 20 times, with final results reported as means with 95% confidence intervals. For all baselines, we replace the original text encoder with `T5-Base`. For MoMask, we implement a 6-layer RVQ. Please refer to supplementary materials for baseline implementations.

**Implementation Details.** Our VQVAE encoder and decoder consists of three dilated res-blocks, with a down(up)-scale factor of 4 [37, 9]. The temporal quantization scales follows the progression $[n/2^V, ..., n/2^0]$ with $n$ denoting the full-scale length. We employ 4 (i.e., $V = 3$) quantization layers for HumanML3D and 2 for `SnapMoGen`, with codebook sizes of $512 \times 512$ and $2048 \times 512$, respectively. The hyper-parameters $\beta$ and $\lambda_{ess}$ are set to 0.02 and 2.0. Our transformer architecture comprise 8 layers with feedforward size of 1024, latent dimension of 384, 6 attention heads, and a dropout ratio of 0.2, totaling 13.5M parameters for *in-context* model, and 18.3M parameters for *cross-attention* model. During inference, we use classifier-free guidance scales of 5 and 4, and iteration counts ($L$) of 10 and 18 for `SnapMoGen` and HumanML3D, respectively. All models are trained on a single Tesla V100 GPU, with batch size of 256 for VQVAEs and 64 for transformers.

## 5.1 Comparison to State-of-the-art Approaches

The quantitative results on HumanML3D and `SnapMoGen` are reported in Tables 2 and 3, respectively. Overall, MoMask++ attains state-of-the-art performance on both datasets, demonstrating consistent improvements in motion-text alignment and motion quality. These advantages are particularly pronounced in our `SnapMoGen` dataset, partially due to the more expressive evaluation model. We observe that previous works struggle with the complex, lengthy text inputs in `SnapMoGen`, and fall short in maintaining multimodal semantic coherence, as evidenced by the relatively low CLIP scores and R-precision values. Notably, our method outperforms MoMask with only two VQ layers (a quarter of MoMask's token count) with similar model size. Between the two variants of MoMask++, we find that the *in-context* model generally performs better on HumanML3D. It however tends to overfit on long text prompts in `SnapMoGen` (Fig. 6) and underperforms compared to the *cross-attention* model. Nevertheless, a significant gap to real motions still exists, suggesting substantial room for future improvements.

Figure 4 displays pose sequences generated by MoMask++, demonstrating its ability to produce precise motions following fine-grained text prompts. We further showcase the capability to handle out-of-domain user prompts by employing an LLM to rephrase the inputs. For additional generation results and comprehensive visual comparisons, please refer to the supplementary materials.

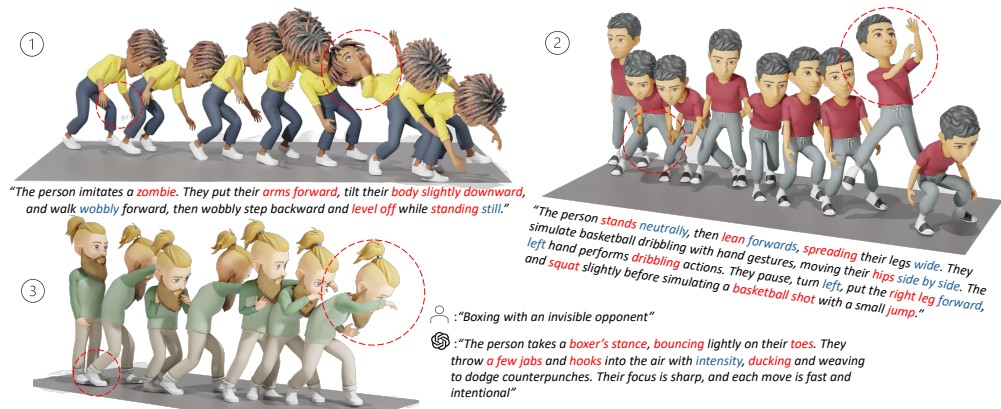

Figure 4: **MoMask++ generated samples** for SnapMoGen test prompts (#1,2) and a casual user prompt (#3).

| | Pose Dim. | VQ Config. | | | | VQ Reconstruction | | T2M Generation | |
|---|---|---|---|---|---|---|---|---|---|
| | | #Codes | #Quant. | F/M | w/ Att. | FID ↓ | Joint Pos. Err. ↓ | FID↓ | CLIP Score ↑ |
| base | 296 | 2048 | 4 | M | ✓ | 2.80 | 8.13 | 15.94 | 0.673 |
| (A) Only essential pose feat. | 148 | 1024 | | | | 4.71 | 7.12 | 15.95 | 0.667 |
| (B) Smaller codebook | | 1024 | | | | 3.30 | 8.43 | **15.61** | **0.668** |
| | | 512 | | | | 3.77 | 8.95 | 16.96 | 0.665 |
| (C) Varying #quant | | | 5 | | | **2.31** | 7.65 | 16.38 | 0.663 |
| | | | 3 | | | 3.13 | 8.40 | 16.21 | 0.652 |
| | | | 2 | | | 4.57 | 8.89 | 15.56 | 0.684 |
| | | | 1 | | | 8.81 | 10.48 | 16.25 | 0.677 |
| (D) Full-scale v.s. multi-scale | | | | F | | 2.64 | **6.53** | 18.02 | 0.667 |
| (E) W/o attention | | | | | ✗ | 3.39 | 8.57 | 16.18 | 0.662 |

Table 4: **Ablation analysis of VQ configuration on SnapMoGen test set.** `"F/M"` denotes full-scale versus multi-scale residual quantization. For text2motion, this experiment use 284 latent dimension, 1024 forward size, and 8 attention layers, with T5-base text encoder and in-context conditioning.

## 5.2 Component Analysis

We perform comprehensive ablation experiments to evaluate the effects of various hyper-parameters and technical designs, as shown in Tab. 4 and Tab. 5. In Tab. 4 (A), we observe that compact pose representation leads to a smaller VQ reconstruction error, while it slightly underperforms for text-to-motion synthesis.

In terms of **VQ configuration**, we observe from Tab. 4 (B) that while increasing codebook size consistently enhances VQ reconstruction and text-motion alignment (CLIP score), motion quality does not necessarily follow this trend (best FID at $|\mathcal{C}| = 1024$). Tab. 4 (C) show that additional VQ layers effectively improve reconstruction, but more token hierarchies also introduce complexity for text-to-motion synthesis, with optimal results at 2 layers. In Tab. 4 (D), we apply full-scale for all layers, yet despite achieving better VQ performance, inefficient token utilization leads to suboptimal generation quality. Finally, incorporating self-attention layers in the encoder and decoder (Tab. 4 (E)) improves both VQ learning and motion synthesis performance.

We then examine the effects of **text augmentation** and **text-to-motion transformer design** in Tab. 5. In Tab. 5 (A), caption augmentation clearly improves model performance across all evaluation metrics. In Tab. 5 (B), we observe that the CLIP text encoder is inadequate for handling the long and complex textual descriptions in `SnapMoGen`. From Tab. 5 and Fig. 6, we further find that cross-attention conditioning is less prone to overfitting and leads to higher motion quality and better text–motion alignment. Meanwhile, Tab. 5 (D) show that transformers with higher latent dimensions or more attention layers counterintuitively degrade motion generation quality. Figure 6 provides additional insight, indicating that larger transformer models (Base: 18.3M, Medium: 36.6M, Large: 53.4M) tend to overfit the dataset more severely.

**How are tokens at different scales modeled?** To investigate this question, we analyze which tokens are "favored" by MoMask++. During iterative inference, MoMask++ generates a complete

| | Text Aug. | T2M Config. | | | T2M Generation | |
|---|---|---|---|---|---|---|
| | | Text Enc. | Conditioning | Architecture | FID↓ | CLIP Score ↑ |
| base | ✓ | T5-base | In-context | (B) 384, 1024, 8 | 15.56 | 0.684 |
| (A) W/o text aug. | ✗ | | | | 17.98 | 0.656 |
| (B) CLIP text enc. | | CLIP | | | 19.96 | 0.478 |
| (C) Cross-att cond. | | | Cross-att. | | **15.06** | **0.685** |
| (D) Larger model | | | Cross-att. | (M) 512, 2048, 8 | 15.58 | 0.679 |
| | | | Cross-att. | (L) 512, 2048, 12 | 16.02 | 0.670 |

Table 5: **Ablation analysis of T2M model configuration on SnapMoGen test set.** `"Architecture"` refers to transformer hyperparameters including latent dimension, feedforward size, and number of layers. This experiment use 2 quantization layers for VQ, with 2048 codebook size and 296 pose features.

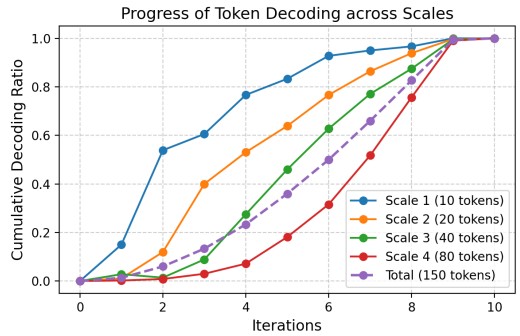

Figure 5: Decoding progress over iterations for different token scales.

Figure 6: Validation loss curves of different model sizes on SnapMoGen.

motion sequence by selectively retaining and re-masking tokens at each step, allowing us to track which tokens the model prioritizes over time. We conduct an experiment using four token scales (from coarse to fine, with 10, 20, 40, and 80 tokens, totaling 150 tokens per sequence) and a 10-iteration inference process over 32 text prompts, recording the token completion ratio at each scale. The results, shown in Fig. 5, reveal that the model naturally prioritizes coarse-scale tokens (Scale 1) in the early stages of generation and progressively shifts its focus toward finer scales. This behavior demonstrates a "global-to-local" generation strategy, indicating that the attention mechanism effectively captures and prioritizes information based on semantic importance (coarse-to-fine).

# 6 Conclusion

In this paper, we introduced SnapMoGen, a high-quality text-motion dataset featuring temporally continuous motion segments with expressive textual annotations. Comprising 20K motion clips and 122K detailed descriptions averaging 48 words each, SnapMoGen provides significantly richer semantic information than existing datasets. We also proposed MoMask++, a novel text-to-motion generation framework that employs multi-scale residual vector quantization and a single generative masked transformer for token prediction. Extensive experiments on both HumanML3D and SnapMoGen demonstrate the state-of-the-art performance of MoMask++.

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

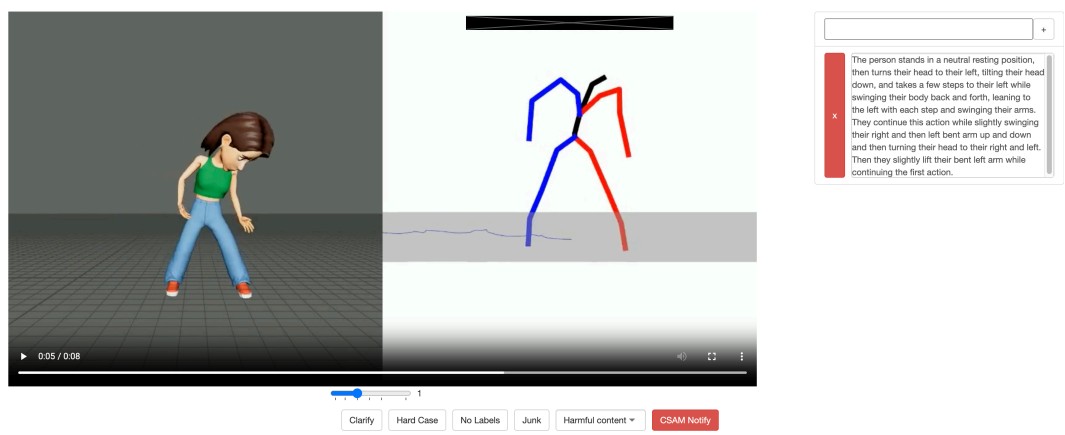

Figure 7: Text annotation UI for `SnapMoGen`.

# A  Annotation Details

Our text annotation interface is presented in Fig. 7. We first visualize all motions using a 3D character to help annotators better understand the motion content. However, since the character may walk out of the camera view and inter-penetration artifacts sometimes occur, we also display the motions using stick-figure representations that remain centered in the camera view. These two visualizations are synchronized and presented simultaneously to annotators. Annotators can also flag low-quality motions during the annotation process.

# B  Evaluation Details

## B.1  Baseline Implementation.

All baseline models on `SnapMoGen` dataset leverage the `T5-base` model for extracting word-level features from text descriptions and are trained using a single NVIDIA RTX A6000 GPU.

For MDM [34], we use an 8-layer transformer decoder where the text encoding is injected via cross-attention layers. The model is trained for 600K steps with a batch size of 1024 using a diffusion process with $T = 1000$ steps. For T2M-GPT [37], we first learn a codebook size of $1024 \times 512$ with a downsampling rate of 4. Then, we model a sequence of codebook indices via an 18-layer transformer. During training, text embeddings and motions are concatenated and processed as input, and a random portion of the ground-truth code indices is replaced with random ones to improve robustness. The model is trained for 600K steps with a batch size of 128. For StableMoFusion [14], we use a Conv1D-based U-Net incorporating residual cross-attention to align motion features with word-level semantics, along with group normalization. The model is trained for 500K iterations with $T = 1000$ denoising steps and a batch size of 1024. For MARDM [23], we first encode motion into a latent representation using a 3-layer ResNet-based auto-encoder. These motion latents are then modeled using a masked autoregressive transformer with a dimension of 1024 and 16 attention heads, where text encodings are injected via cross-attention layers. The model is trained for 600K steps with a batch size of 128.

## B.2  Evaluation Model

Our evaluation model accounts for both motion fidelity and text-motion alignment. We adopt the TMR framework, as shown in Figure 8. This framework comprises three network components: a motion encoder that encodes motion sequences into global vectors, a text encoder that encodes text sequences into global vectors, and a motion decoder that reconstructs motions from either motion or text vectors. All three networks are 6-layer transformers with a latent dimension of 256, 4 attention heads, and a feedforward hidden size of 1024. The `T5-base` model first extracts word-level features from texts. For motions, we use only the essential 148-dimensional root motion and local rotational features. All encoders output Gaussian distribution parameters (mean and log-variance), from which

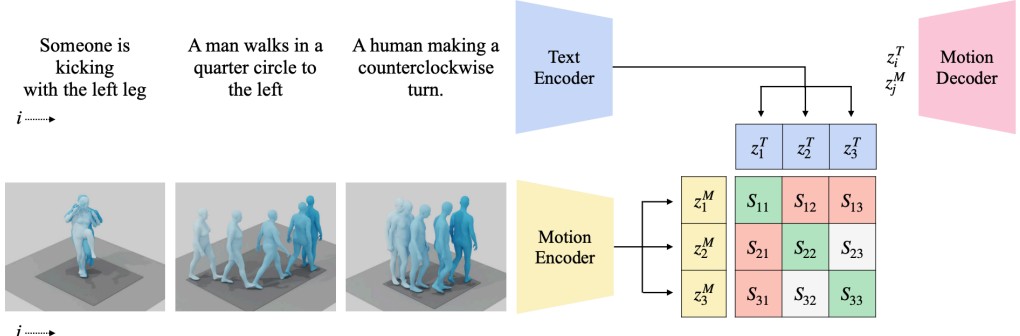

Figure 8: **Architecture of the evaluation model** [26]. Three network components are trained with two main goals: multimodal alignment and reconstruction. The cosine similarity between motion embeddings and text embeddings from positive pairs (green) is maximized, while similarity for negative pairs is minimized. Meanwhile, both embeddings are required to reconstruct the corresponding motion sequence through the motion decoder. Image adapted from TMR [26].

vectors are sampled. We append two temporal timesteps at the end of the sequence input for outputing these vectors.

This evaluation model is trained with a compound loss:

$$\mathcal{L}_{\text{tmr}} = \mathcal{L}_{\text{rec}} + \lambda_{\text{KL}}\mathcal{L}_{\text{KL}} + \lambda_{\text{E}}\mathcal{L}_{\text{E}} + \lambda_{\text{NCE}}\mathcal{L}_{\text{NCE}},$$

where $\mathcal{L}_{\text{rec}}$ measures the motion reconstruction given text or motion input (via a smooth L1 loss). A KL-divergence loss $\mathcal{L}_{\text{KL}}$ regularizes each embedding distribution to be close to a unitary Gaussian distribution $\mathcal{N}(\mathbf{0}, \mathbf{I})$, and also encourage these two distributions to be close to each other. $\mathcal{L}_{\text{E}}$ enforces both mean vectors to be similar to each other. Finally, a InfoNCE [24] loss is used for constrastive learning of motion-text batches with batch size of 64. We set $\lambda_{\text{E}}$, $\lambda_{\text{KL}}$, and $\lambda_{\text{NCE}}$ to $1e-5$, $1e-5$, and $0.1$. For more model details, we recommend to read the original TMR work [26].

In inference, we employ the evaluation metrics designed in [10]. We increase the pool size for R-precision to 100 and directly use the mean vectors of the latent distributions as embedding vectors

## C   LLM-based Prompt Augmentation

**Dataset Augmentation.**   During training, we enhance data diversity by employing an LLM to rewrite human-provided annotations, generating paraphrased versions with varied linguistic structures while preserving core semantics. This approach ensures each motion sequence is associated with multiple textual descriptions, improving model robustness. The prompt instructions for ChatGPT are provided in Tab. 6.

**Inference-time Prompt Augmentation.**   During inference, the LLM rewrites each input prompt into a richly detailed description, incorporating explicit motion cues such as body posture, timing, and stylistic elements. This expanded form more effectively guides motion generation models. The instructions for ChatGPT are provided in Tab. 7.

## D   Limitation

We present several representative failure motions in the static webpage. Here we discuss limitations from both data and model perspectives.

**Dataset.**   Despite extensive calibration and post-processing of the collected motions, quality issues rooted in the inertial-based mocap suit persist. For example, global positions may lack precision, and jitters can occur during fast or complex motions. Additionally, we are unable to capture highly skilled motions such as cartwheels, backflips, or outdoor activities (e.g., climbing).

**Task Description** The goal is to rewrite textual descriptions from a text-to-motion dataset to correct typos and grammar while rephrasing them for better readability and fluency. The key requirement is that the semantics of the described human motion must be **preserved exactly**, with no loss or modification of motion detail. You may vary the sentence structure, use synonyms, merge or split phrases, or improve flow, but **the described body movements and temporal order must remain unchanged**.

**Instructions**

1. Correct all spelling, grammar, and stylistic issues in the input text.

2. Rephrase the input to make it clearer, more fluent, and more readable.

3. You **must not** remove, simplify, or alter the described body movements.

4. Keep all motions, ordering, and temporal logic intact.

5. Add mild clarifications only if they help with motion clarity.

**Examples**

*Original:* The person takes two steps forward, starting with his left foot, bends down and reaches down with his right hand to the floor, rises with his arms out to the sides and steps back, and abruptly takes a step forward with his right foot with his arms bent at the elbows, and shaking both arms slowly steps forward standing in a fighting stance and provoking a fight.

*Augmented:* The person steps forward twice, beginning with the left foot. They bend down, reaching the floor with their right hand. Rising, they extend arms sideways and step back. Suddenly, they step forward with the right foot, elbows bent, arms shaking. They stand in a fighting stance, slowly advancing, as if provoking a fight.

*Original:* The person stands with their legs wide apart. Then they take two steps back and slightly to the left, lowering their head down and raising their left hand to their head. Then they lower their left hand and take two steps to the right, stopping. Then walks forward, turning to the left and waving their arms.

*Augmented:* The person stands with legs spread wide apart. They move two steps backward and slightly to the left while lowering their head and raising their left hand to touch it. Afterward, they lower their left hand, take two steps to the right, and pause. Then they walk forward, turning towards the left while waving their arms in a fluid motion.

Table 6: Prompt instruction for grammar-correcting and semantically-preserving text augmentation.

**Model.** Opportunities for improving text-to-motion models also remain. As MoMask++ relies on VQ, quantization errors inevitably degrade motion quality. We observe that MoMask++ struggles with rare motion patterns or uncommon text prompts. Furthermore, it does not yet maintain physical plausibility, such as proper foot contacts.

**Task Description:** Your task is to rewrite text prompts of user inputs for a text-to-motion generation model inference. This model generates 3D human motion data from text, you need to understand the intent of the user input and describe how the human body should move in detail, and give me the proper duration of the motion clip, usually from 4 to 12 seconds.

**Instructions:**

1. Make sure the rewritten prompts describe the human motion without major information loss.

2. Be related to human body movements—the tool is not able to generate anything else.

3. The rewritten prompt should be around 60 words, no more than 100.

4. Use a clear, descriptive, and precise tone.

5. Be creative and make the motion interesting and expressive.

6. Feel free to add physical movement details.

**Examples:**
*Input:* Shooting a basketball.
*Rewrite:* The person stands neutrally, then leans forward, spreading their legs wide. They simulate basketball dribbling with hand gestures, moving their hips side to side. The left hand performs dribbling actions. They pause, turn left, put the right leg forward, and squat slightly before simulating a basketball shot with a small jump.
*Length:* 8 seconds

*Input:* Zombie walk.
*Rewrite:* The person shuffles forward with a stiff, dragging motion, one foot scraping the ground as it moves. His arms hang loosely by its sides, occasionally jerking forward as it staggers with uneven steps.
*Length:* 6 seconds

Table 7: Instructions for re-writing casual user prompts.

# NeurIPS Paper Checklist

