# OpenReview forum: "SnapMoGen: Human Motion Generation from Expressive Texts"
_NeurIPS.cc/2025/Conference — NeurIPS 2025 poster_

### Official Review · Reviewer_zsJy · 2025-06-29

**Clarity:** 3
**Significance:** 3
**Originality:** 2
**Rating:** 4
**Confidence:** 4

**Summary:**

This paper introduces OmniMotion, a new human motion dataset and generation framework. It presents a large scale collection of continuous motion capture sequences paired with rich, paragraph length textual descriptions that cover action details, style, and context. The paper also proposes MoMask++, a multi-scale residual vector quantization and unified masked Transformer approach for tokenization and parallel generation of motion sequences. Experiments demonstrate state-of-the-art performance on benchmarks and ablation studies identify optimal quantization and model configurations.

**Questions:**

(1) The paper relies on LLM‐based caption augmentation, yet the examples suggest mostly simple synonym swaps. Could the authors include an ablation study that compares training with and without these augmented captions to quantify its actual impact on model performance? If such an experiment shows a clear gain in generation quality or retrieval metrics, my confidence in the dataset’s contribution would increase.

(2) During inference, prompts are likewise expanded via an LLM to match the training style. How do you verify that these expanded prompts faithfully capture the user’s intent without introducing bias? Have you measured whether the added detail reduces motion diversity? Providing some visualization on diversity vs. prompt length would help clarify the model’s robustness and could strengthen the paper.

**Ethical Concerns:**

["NO or VERY MINOR ethics concerns only"]

**Final Justification:**

I thank the authors for their responses. In the rebuttal, the authors used experimental results to demonstrate that their caption augmentation can improve performance, which shows the value of their dataset. Overall, I think this paper proposes a valuable dataset, with relatively comprehensive experiments and good writing. However, using LLMs to enhance text captions is a standardized practice, and the method itself is an incremental engineering effort. Therefore, I tend to maintain my original rating.

**Limitations:**

Please add a discussion of the limitations, for example, whether LLM-annotated text is truly expressive or merely a longer, semantically equivalent restatement.

**Quality:**

3

**Strengths And Weaknesses:**

Strengths:
(1) The paper delivers a large-scale, richly annotated human motion dataset.
(2) Proposed MoMask++ model that fuses multi scale RVQ and a unified masked Transformer to reduce token usage while improving reconstruction quality.
(3) Leverages LLM-based prompt augmentation both to enrich dataset annotations and to harmonize inference inputs with training style.
(4) Presents extensive benchmark comparisons and ablation studies that clearly demonstrate the dataset’s value and the model’s performance gains.

Weaknesses:
(1) During dataset generation, augmented captions rely mostly on simple synonym swaps and lack deeper structural or semantic variation
(2) The shift from MoMask to RVQ, while effective, represents a modest, incremental improvement over existing quantization approaches.
(3) Inference appears to depend on LLM-expanded, detailed prompts—raising questions about the model’s ability to generate diversity results.

---

> ### Author Rebuttal · Authors · 2025-07-28
>
> **We appreciate your time and effort spent on reviewing our paper. Below, we answer the raised questions one by one.**
>
> ***Q1**:  Could the authors include an ablation study that compares training with and without these augmented captions to quantify its actual impact on model performance? If such an experiment shows a clear gain in generation quality or retrieval metrics, my confidence in the dataset’s contribution would increase.*
>
> **A1**: Thank you for your valuable suggestion. Our LLM-based caption augmentation specifically aims to increase textual variation without altering or missing core motion semantics. We conducted an ablation study comparing our model trained with and without these augmented captions, with results below:
>
> |    | FID$\downarrow$ |  Top1$\uparrow$  |  Top2$\uparrow$  |  Top3$\uparrow$  |  CLIP Score$\uparrow$  |
> |:--:|:--:|:--:|:--:|:--:|:--:|
> |  Ours  |  15.56|  0.802  |  0.905  |  0.938   |  0.684  |
> |  Ours w/o text aug  |  17.98  |  0.758  |  0.873  |  0.917  |  0.656  |
>
> The table clearly shows that caption augmentation significantly enhances model performance across all metrics. Our training curves also show that text augmentation makes the model less prone to overfitting.
>
> ---
>
> ***Q2**: During inference, prompts are likewise expanded via an LLM to match the training style. How do you verify that these expanded prompts faithfully capture the user’s intent without introducing bias?*
>
> **A2**: During inference, we instruct the LLM with descriptive prompts and in-context examples (Table 6 in supplementary) to "understand the user's intent and rewrite a clear, descriptive, precise-tone prompt without major information loss."
>
> Our primary verification involved **meticulous manual checks on hundreds of samples**, comparing the original user prompt, the rewritten prompt, and the resulting generated motion. Given the capabilities of current LLMs for this specific task, we found that the rewritten prompts and their generated motions reliably and precisely reflect the original user intent.
>
> Nevertheless, we acknowledge that potential risks may still exist, and we are happy to discuss this in the limitations and outlook sections.
>
> ---
>
> ***Q3**: Have you measured whether the added detail reduces motion diversity? Providing some visualization on diversity vs. prompt length would help clarify the model’s robustness and could strengthen the paper.*
>
> **A3**: Thank you for the suggestion. We have not measured the effect of added detail on motion diversity, as it’s difficult to be quantitatively evaluated. Intuitively, more detailed prompts reduce ambiguity and naturally constrain diversity. We agree that qualitative examples would help clarify this trade-off and will include them in the revision. It's worth noting that many raw user prompts (e.g., “walk like a robot”, “bird taking flight”) fail without added detail, as shown in our ablation videos.
>
> ---
> ***Q4**: Please add a discussion of the limitations, for example, whether LLM-annotated text is truly expressive or merely a longer, semantically equivalent restatement.*
>
> **A4**: Thanks! We will add this to our limitation discussion.
>
> ---
> ***Q5**: The shift from MoMask to RVQ, while effective, represents a modest, incremental improvement over existing quantization approaches.*
>
> **A4**: Thank you for the comment. Our multi-scale motion quantization improves over MoMask's RVQ by achieving comparable reconstruction with only one third of token counts. In motion generation, MoMask++ notably outperforms MoMask on complex prompts in OmniMotion, reducing FID from 17.4 to 15.06. As shown in our supplementary videos, MoMask++ better handles fine-grained control (e.g., “right leg in front”) and rare actions (e.g., “figure skater spinning”). Nevertheless, we acknowledge that there are still much room to improve in future.
>
> ---
> Thanks again for your comment. We sincerely hope our responses have properly addressed your concerns.

---

> > ### Comment · Reviewer_zsJy · 2025-08-05
> >
> > I thank the authors for their responses. In the rebuttal, the authors used experimental results to demonstrate that their caption augmentation can improve performance, which shows the value of their dataset. Overall, I think this paper proposes a valuable dataset, with relatively comprehensive experiments and good writing. However, using LLMs to enhance text captions is a standardized practice, and the method itself is an incremental engineering effort. Therefore, I tend to maintain my original rating.

---

> > > ### Author Response · Authors · 2025-08-06
> > >
> > > Thanks for your reply.

---

### Official Review · Reviewer_bxuM · 2025-07-02

**Clarity:** 3
**Significance:** 3
**Originality:** 3
**Rating:** 4
**Confidence:** 4

**Summary:**

This paper tackles the problem of human motion synthesis from text. The authors propose a new motion dataset called OmniMotion which consists of around 20k motion sequence each with 6 detailed textual description. Different from preivous motion datasets whose text description is brief and general, OmniMotion provides highly expressive text descriptions for each motion enables the potential of fine-grain control in motion synthesis. Building upon the SOTA method MoMask, the authors further introduce MoMask++. The main contribution of MoMask++ is the multi-scale motion quantization which enables better motion reconstruction with less tokens.

**Questions:**

L202, how does the CFG used in an auto-regressive model?

**Ethical Concerns:**

["NO or VERY MINOR ethics concerns only"]

**Final Justification:**

I would suggest accept this paper. The reubuttal addressed most of my major concerns. The authors provide a clarification of the contribution between their method and MoMask++ and detailed discussion about the performance on HumanML3D dataset.

**Limitations:**

yes

**Quality:**

3

**Strengths And Weaknesses:**

**Strengths**
- The overall quality of the writting is good. The arguments are clear and valid.
- The motivations of the new dataset as well as the multi-scale motion quantization are convincing.
- The highly detailed text description provided by the new dataset can be very helpful to the future reseach of text-to-motion.

**Weakness**
- The technical contribution is limited. The proposed MoMask++ is similar to the MoMask pipeline with differences on residual quantization and text encoding. And the only contribution is the multi-scale residual quantization. There is no novelty on the text encoding since it simply uses an existing model.
- The performance of the method on HumanML3D dataset is either marginally better or even worse comparing to MoMask. While on OmniMotion dataset, the proposed method is consistently better. Are there any explainations on this? Is it because the T5-based text encoder better leverages the highly detailed descrition of OmniMotion dataset? If so, what is the benefits of the multi-scale residual quantization in text-to-motion?

---

> ### Author Rebuttal · Authors · 2025-07-28
>
> **We thank you for your thoughtful comments. The following responses aim to address your concerns point by point.**
>
> ***Q1**: The proposed MoMask++ is similar to the MoMask pipeline with differences on residual quantization and text encoding. And the only contribution is the multi-scale residual quantization.*
>
> **A1**: Besides multi-scale residual quantization, we also employ an **unified generative transformer** which allows more **flexible token prioritization**. Unlike MoMask, which uses separate models for different scales and follows a fixed, layer-by-layer generation schedule, MoMask++ employs a single, universal Transformer to generate tokens across all scales. This design is not only more efficient but crucially enables the model to **dynamically and flexibly prioritize token generation** at any scale during iterative inference. As detailed in our response to Reviewer xbdR (Q3), MoMask++ naturally learns a "global-to-local" strategy, prioritizing coarse-scale tokens early while also selectively valuing important finer-scale tokens.
>
> **LLM-Enhanced Generalizability**: Our novel integration of LLMs to rephrase casual, out-of-domain user prompts into detailed, expressive descriptions is a critical technical contribution. This approach, uniquely enabled by our OmniMotion dataset's expressive annotations, allows MoMask++ to achieve unprecedented generation on novel, complex prompts like "a bird taking flight" or "chasing after a fluttering butterfly." This is a practical advancement for real-world text-to-motion applications.
>
> ---
>
> ***Q2**: The performance of the method on HumanML3D dataset is either marginally better or even worse comparing to MoMask. While on OmniMotion dataset, the proposed method is consistently better. Are there any explanations on this? Is it because the T5-based text encoder better leverages the highly detailed description of OmniMotion dataset? If so, what is the benefits of the multi-scale residual quantization in text-to-motion?*
>
> **A2**: First, to clarify, all baselines, including MoMask, were adapted to use the **T5 text encoder for fair comparison** on OmniMotion (an ablation is provided in Table 4, F Row). The performance difference is not due to a text encoder advantage.
>
> The disparity arises from the nature of the datasets: HumanML3D typically features simple text semantics, where MoMask performs well. In contrast, OmniMotion's **much longer and more complex texts** critically stress a model's capacity for fine-grained text-motion mapping. MoMask++ is uniquely advantaged on OmniMotion due to:
>
> 1. **Compact, Detail-Preserving Multi-Scale VQ**: Our multi-scale VQ design effectively preserves motion details while yielding significantly more compact token sets (e.g., 45% fewer tokens than MoMask Residual VQ). This eases the burden of generating extensive tokens from long, complex text prompts.
>
> 2. **Token Exploitation**: By confining temporal scales at each quantization layer, our VQ encourages tokens to continuously capture meaningful semantics hierarchically (as shown in Figure 3). This contrasts with MoMask's VQ, where first layer is overly dominant and latter layers are often "lazy" to capture incremental information. Our design's efficient information encoding facilitates superior text details to motion mapping.
>
> 3. **Flexible, Single-Stage Generative Transformer**: Unlike MoMask's rigid, fixed-order two-stage generation, our one-stage masked Transformer dynamically and flexibly prioritizes token generation across all scales during iterative decoding (as elaborated in our A1 to Reviewer xbdR (Q3)).
>
> Collectively, these advantages make MoMask++ a better candidate for coping with the long and complex texts found in OmniMotion.
>
> ---
>
> ***Q3**: L202, how is the CFG used in an auto-regressive model?*
>
> **A3**: We'd like to clarify that **MoMask++ is not an auto-regressive model**. Instead, it is a non-autoregressive masked generative model where **all tokens are predicted in parallel** at each iteration and then re-masked for subsequent iterations.
>
> Consistent with the MoMask approach, Classifier-Free Guidance (CFG) is applied during inference at the final linear projection layer before softmax. The guided logits $ω_g$ are computed by: $ω_g =(1+s)⋅ω_c −s⋅ω_u$ where $ω_c$ are the conditional logits, $ω_u$ are the unconditional logits, and $s$ is the guidance scale.
>
> ---
> We sincerely hope that we have properly addressed your concerns. If not, we are happy to open further discussions.

---

> > ### Comment · Reviewer_bxuM · 2025-08-05
> > **responses to the rebuttal**
> >
> > I would like to thank the authors for the rebuttal which has addressed most of my concerns. Thus, I would like to maintain my original score to suggest to accept this paper.

---

> > > ### Author Response · Authors · 2025-08-06
> > >
> > > Thanks for your reply.

---

### Official Review · Reviewer_PaWz · 2025-07-02

**Clarity:** 3
**Significance:** 2
**Originality:** 2
**Rating:** 4
**Confidence:** 4

**Summary:**

This paper introduces MoMask++, a multi-scale masked transformer for text-to-motion generation, alongside the OmniMotion dataset with detailed text descriptions. While the work shows solid technical execution and experimental effort, several concerns regarding novelty and methodology limit its contribution.

**Questions:**

Questions to Address

Novelty: How does your multi-scale approach fundamentally differ from prior works like MoFusion or T2M-GPT, beyond standard cross-scale attention? Can you better justify the novelty given the modest performance improvement over MoMask?

Multi-Scale Fusion: Why was summation chosen for fusing multi-scale features? Have you explored alternative, weighted fusion methods to avoid over-emphasizing low-frequency information? What analysis can you provide on the post-fusion features?

Attention & Positional Encoding: How does the attention mechanism handle concatenated sequences of very different lengths (e.g., 2 vs. 80 tokens)? How do positional encodings function after this multi-scale concatenation to preserve temporal order?

Dataset Timestamps: Could you enhance the dataset by aligning text descriptions with timestamps to allow for motion speed and pace control?

Motion Quality: How do you plan to address the foot-ground contact issues mentioned in the limitations?

Visualization: Can you revise Figure 2 to make the visualizations for "Motion VQ code" and "Motion embedding" more distinct to avoid confusion?

**Ethical Concerns:**

["NO or VERY MINOR ethics concerns only"]

**Final Justification:**

Almost all reviewers have expressed similar views, acknowledged the contributions of this paper, and agreed that it falls within the borderline acceptance range.

**Limitations:**

Yes

**Quality:**

2

**Strengths And Weaknesses:**

Strengths:
1. The OmniMotion dataset with extended text descriptions fills a practical gap and could benefit the community.
2. The paper is well-structured and easy to follow, with comprehensive experiments and useful ablation studies.
3. The LLM-based prompt rewriting feature shows thoughtful consideration for real-world usability.
4. The multi-scale residual quantization framework demonstrates some improvements.

Main Concerns:
1. Limited Novelty. This is the main concern with the paper. The core contribution appears to be primarily an engineering combination of existing approaches. While the multi-scale extension of MoMask is reasonable, similar multi-scale strategies have been widely explored in computer vision and motion generation (e.g. MoFusion, T2M-GPT) and cross-scale attention is fairly standard. The improvements shown in Tables 2 and 3 are relatively modest compared to MoMask, which doesn't strongly validate the method's effectiveness.

2. Methods.
a. The paper employs summation for multi-scale information fusion, but it remains unclear whether this is the optimal approach. The use of equal-weight summation may lead to overemphasis of low-frequency information (global patterns) while underweighting high-frequency information (local details). A more detailed analysis of the post-fusion features is needed.
b. When concatenating sequences of very different lengths (e.g., 2 vs 80 tokens), the attention mechanism might naturally favor longer sequences.  Some discussion of this issue would strengthen the paper. Also, It's unclear how positional encodings work after multi-scale concatenation, which seems important for preserving temporal relationships.

3. Dataset: Figure 1 shows the text annotations for motion, which include a series of continuous action descriptions. However, these are merely descriptions of consecutive sub-action contents, lacking alignment with time steps. Therefore, we cannot determine the speed or pace of each sub-action from these descriptions. I believe supplementing this information would be beneficial for text-to-motion generation.

4. Minor Issues
a. The foot-ground contact issues mentioned in limitations suggest room for improvement in basic motion quality.
b. In Figure 2, the Motion VQ code in (a) and the Motion embedding in (c) look too similar, which could cause confusion. A more distinguishable visualization approach should be used.

---

> ### Author Rebuttal · Authors · 2025-07-27
>
> **We thank the reviewer for the comments, and hope the following response properly addresses your concerns.**
>
> ***Q1**: Novelty: While the multi-scale extension of MoMask is reasonable, similar multi-scale strategies have been widely explored in computer vision and motion generation (e.g. MoFusion, T2M-GPT). Can you better justify the novelty given the modest performance improvement over MoMask?*
>
> **A1**: Thanks for your feedback. We wish to clarify that our work offers distinct innovations **beyond a mere engineering combination**, specifically in:
>
> 1. **Novel Multi-Scale Token Design for Motion**: We respectfully clarify that neither MoFusion nor T2M-GPT employs a multi-scale token design as ours does. MoFusion is a continuous diffusion model, and T2M-GPT uses single-layer tokens. To the best of our knowledge, we are the **first** to introduce this multi-scale tokenization for motion generation, enabling each layer to capture distinct, meaningful semantics (Fig. 3).
>
> 2. **Compact & Flexible Token Modeling**: Our design models motion tokens significantly more efficient than MoMask's: achieving the same VQ reconstruction with **2/3 fewer** tokens and requiring **3/4 fewer** tokens for generation. Crucially, a single generative transformer models all tokens, allowing the model to adaptively prioritize important tokens across **all scales** during inference, unlike MoMask's fixed layer order.
>
> 3. **Demonstrable Gains on Complex Prompts**: While HumanML3D metrics are overall modest, our core strength is handling long, complex text prompts. On OmniMotion, MoMask++ shows notable improvements over MoMask (FID 17.4 to 15.06; CLIP 0.66 to 0.685). Qualitatively, our model demonstrates superior understanding of nuanced semantics (e.g., "right leg in front") and complex motions (e.g., "figure skater spinning") that MoMask struggles with (see supp. files).
>
> Finally, we encourage a **holistic** evaluation of our contribution. Our LLM-based prompt rewriting method, made possible by our unique **OmniMotion dataset**, further enhances the model's practical utility to handle novel and complex user prompts (e.g., "bird flight," "frog jump"), beyond what existing methods can achieve.
>
> ---
>
> ***Q2**: Why was summation chosen for fusing multi-scale features? Have you explored alternative, weighted fusion methods to avoid over-emphasizing low-frequency information? What analysis can you provide on the post-fusion features?*
>
> **A2**: **Equal-weight summation is mathematically necessary** for our VQ learning framework. Specifically, we decompose the input feature $f$ into hierarchical residuals across temporal resolutions, and at each resolution the residual features are quantized. For the final summed feature $\hat{f}$ to precisely approximate the input $f$, the quantized features from all levels must be summed **uniformly**, being consistent with the *residual decomposition* process.
>
> Our design inherently prevents over-emphasis on low-frequency information:
> * **Early quantization layers** (i.e., global pattern) are explicitly assigned with shorter token sequences.
> * **Later layers** capture high-frequency local details with more tokens.
> ---
>
> ***Q3**: When concatenating sequences of very different lengths (e.g., 2 vs 80 tokens), the attention mechanism might naturally favor longer sequences.  How does the attention mechanism handle concatenated sequences of very different lengths (e.g., 2 vs. 80 tokens)?*
>
> **A3**: Thanks for this very insightful comment. To investigate this question, we look into the tokens *"favored"* by MoMask++. During iterative inference, MoMask++ generates a complete sequence by retaining and re-masking certain tokens. This allows us to track which tokens the model prioritizes. We conducted an experiment with four token scales and 10-iteration inference using 32 text prompts, tracking the **token completion ratio** at each scale. The results are shown below:
> | Iterations | Scale 1 (10 tokens) |  Scale 2 (20 tokens)  |  Scale 3  (40 tokens)  |   Scale 4 (80 tokens)  |   Total (150 tokens) |
> |:----:|:----:|:----:|:----:|:----:|:----:|
> |Iter 0|0.0|0.0|0.0|0.0|0.0|
> |Iter 1|0.1500|0.0111|0.0280|0.0021|0.0133|
> |Iter 2 |0.5389|0.1194|0.0139|0.0083|0.0600|
> |Iter 3|0.6056|0.4000|0.0889|0.0299|0.1333|
> |Iter 4 |0.7667|0.5306|0.2750|0.0715|0.2333|
> |Iter 5 |0.8333|0.6389|0.4597|0.1813|0.3600|
> |Iter 6 |0.9278|0.7667|0.6278|0.3160|0.5000|
> |Iter 7|0.9500|0.8639|0.7708|0.5174|0.6600|
> |Iter 8|0.9667|0.9389|0.8750|0.7569|0.8267|
> |Iter 9 |1.0|0.9972|0.9958|0.9903|0.9933|
> |Iter 10 |1.0|1.0|1.0|1.0|1.0|
>
> As the table shows, despite their shorter length, the model naturally prioritizes coarse-scale tokens (Scale 1) early in the generation process. It then gradually shifts its focus to finer-scale tokens, demonstrating a "global-to-local" generation strategy. This indicates that the attention mechanism effectively captures and prioritizes information based on its semantic importance (coarse-to-fine) rather than simply token length.
>
> We will include this in our revision as suggested.
>
> ---
>
> ***Q4** : It's unclear how positional encodings work after multi-scale concatenation, which seems important for preserving temporal relationships.*
>
> **A4**: First, each token sequence at every scale is padded to a **fixed maximum length** before concatenation. Then, all token sequences are concatenated by scale (e.g., Scale 1 tokens, then Scale 2, etc.), forming a single long sequence. This resulting sequence is then **incrementally positional encoded** as a standard sequential input. The model recognizes a token's scale from its fixed positional region within this concatenated sequence, and temporal order is also maintained within each scale's token sequence.
>
> We also explored a "separate encoding" strategy, where tokens within each scale were separately PE'd and added with a scale-specific embedding before concatenation. However, this approach led to worse performance (FID increased from 15.06 to 16.45, CLIP score dropped from 0.685 to 0.670).
>
> ---
>
> ***Q5**: Dataset Timestamps: Could you enhance the dataset by aligning text descriptions with timestamps to allow for motion speed and pace control?*
>
> **A5**: Thank you for the suggestion. In OmniMotion, each motion clip is already associated with **precise start and end frame timestamps**, which are encoded directly in the clip filenames. For example, clips named *ep1_00000#0#281* and *ep1_00000#281#503* indicate consecutive segments spanning frames 0–281 and 281–503, respectively.
>
> These timestamps enable downstream applications to control motion speed and temporal alignment. We encourage the reviewer to refer to the provided *Sample_annotations.json* file for more examples and details.
>
> ---
>
> ***Q6**: Motion Quality: How do you plan to address the foot-ground contact issues mentioned in the limitations?*
>
> **A6**: We agree that foot-ground contact is a critical aspect of motion realism. The issue typically stems from two factors:
>
> 1. **Mismatch between global trajectory and local motion** (e.g., the character appears to slide or move unnaturally fast/slow), and
> 2. Local foot jittering or penetration artifacts.
>
> To address these, we see several promising directions:
> 1. For global-local mismatch, recent methods like GenMoStyle [1] and NeMF [2] learn a regressor to infer global motion from local features. A similar approach could be integrated into our framework.
> 2. For local foot artifacts, classical foot inverse kinematics (IK) can help enforce contact constraints.
>
> More generally, integrating **physics-based priors or simulators** could further enhance motion plausibility and enforce realistic contact dynamics.
>
> ---
> ***Q7**: Visualization: Can you revise Figure 2 to make the visualizations for "Motion VQ code" and "Motion embedding" more distinct to avoid confusion?*
>
> **A7**: Thanks for pointing this out. We will revise accordingly.
>
> ---
> Hope our reply satisfactorily addresses your concerns. Otherwise, we will be happy to further discuss.
>
> [1] Generative Human Motion Stylization in Latent Space, ICLR 2024.
> [2] NeMF: Neural Motion Fields for Kinematic Animation, NeurIPS 2022

---

> > ### Comment · Reviewer_PaWz · 2025-08-06
> >
> > Thank you for your rebuttal. I have raised my rating.

---

> > > ### Author Response · Authors · 2025-08-06
> > >
> > > Thanks for your reply.

---

### Official Review · Reviewer_xbdR · 2025-07-03

**Clarity:** 3
**Significance:** 2
**Originality:** 2
**Rating:** 4
**Confidence:** 5

**Summary:**

The authors propose a new text2mtoion dataset and revisit the Momask model to get better results. The dataset is useful and meaningful. It comprises 20K motion clips totaling 44 hours, accompanied by 122 detailed textual descriptions averaging 48 words per description (vs. 12 words of HumanML3D).

**Questions:**

1. How the author handle the halluciation problem when only using the text as the prompt?

**Ethical Concerns:**

["NO or VERY MINOR ethics concerns only"]

**Final Justification:**

The authors solved most of my questions and further clarify their framework contirbution to me. This work is still more like an engineering work to me. I decided to keep my score as a Borderline accept.

**Paper Formatting Concerns:**

The ablation table needs to refine. It is hard to recognize.

**Quality:**

3

**Strengths And Weaknesses:**

Strengths:
1. Dataset is good. The t2m field nees more high quality data.
2. The authors provides extensive experiments results and visualization results, which are not observed before.
3. The evaluation process is standardized and meaningful.

Weaknesses:
1. The paper do not propose a new generation model or architecture, but an incremental improvment on the existing method.
2. The rewriting process is intuitive, since 2/3 of the texts are generated by LLM. And the generated texts have it own bias. I do not think this should be highlighted in the abstract and it is just a trick everybody knows.
3. Although the dataset is useful, the paper do not provide any new insight for me. It is more like an engineering project. Thus, I will give a boarderline this time.

---

> ### Author Rebuttal · Authors · 2025-07-27
>
> **We thank the reviewer for the comments, and hope the following response properly addresses your concerns.**
>
>
> ***Q1***: *The paper does not propose a new generation model or architecture, but an incremental improvement on the existing method. Although the dataset is useful, the paper does not provide any new insight for me. It is more like an engineering project.*
>
> **A1**: Thanks for your comment. While our model builds on existing methods, our paper's core novelty and insight lie in two areas:
>
> First, we highlight the **critical, under-explored role of expressive text input** in text-to-motion generation. Our OmniMotion dataset, with its uniquely rich and detailed annotations, proves that fine-grained text-based control and generalization are possible when models are provided with sufficient textual cues. This enables generating motions from complex prompts like "a bird taking flight," which were previously impossible. We also benchmark a wide range of baseline models on OmniMotion under fair settings.
>
> Second, our MoMask++ model introduces a more efficient multi-scale motion quantization and uses a single transformer, making it a neater and more effective solution. It also demonstrably outperforms previous methods on long text prompts within our challenging OmniMotion dataset, pushing the state of the art in practical application.
>
> These two contributions collectively offer new insights and practical advancements for the field.
>
> ---
>
> ***Q2**: The rewriting process is intuitive, since 2/3 of the texts are generated by LLM. And the generated texts have their own bias. I do not think this should be highlighted in the abstract and it is just a trick everybody knows.*
>
> **A2**: Thanks for the feedback. We want to clarify that our LLM integration is a **strategic enabler**, not just a trick. We use LLMs in two ways:
> 1. **Dataset Curation**: For minor corrections and semantic expansions to ensure OmniMotion's high-quality, expressive annotations. I also use LLM to increase the text variations in the dataset
> 2. **Inference-Time Prompt Enhancement**: Crucially, LLMs enrich casual user prompts with motion details, bridging the gap between typical user input and the detailed motion cues our model requires.
>
> We are highlighting the second point in the abstract to underscore how our expressive dataset enables generalizable motion generation from diverse user queries. We'll revise the abstract to clarify this key application.
>
> ---
>
> ***Q3**: How does the author handle the hallucination problem when only using the text as the prompt?*
>
> **A3**: We assume the reviewer is referring to **LLM hallucination during prompt rewriting**. Please correct us if this is not the case.
> We address this by:
> 1. **Strict Prompting**: We follow best practices from image/video generation [1,2], using highly descriptive prompts to guide the LLM.
> 2. **Task Simplicity**: For dataset augmentation, the LLM's role is limited to correcting typos and rephrasing for diversity while strictly preserving original motion semantics. For inference, it adds motion details without altering user intent.
> 3. **Quality Control**: We provide several in-context examples to reduce ambiguity and have meticulously checked rewriting samples.
>
> Given the **relatively low complexity** of these rewriting tasks, we observed negligible instances of factual errors or "hallucinations" in the LLM-generated prompts (refer to Tables 5 & 6 in the supplementary material). We are happy to discuss this further in the limitations section if the reviewer deems it necessary.
>
> ---
>
> ***Q4**: The ablation table needs to be refined. It is hard to recognize.*
>
> **A4**: Thank you for the suggestion. We will reorganize the ablation table to improve readability in the version.
>
> ---
> We sincerely hope our response addresses all your concerns. Please feel free to let us know if you have further questions.
>
> [1] Betker et al., Improving image generation with better captions. 2023.
> [2] Bao et al., Vidu: a highly consistent, dynamic and skilled text-to-video generator with diffusion models. 2024.

---

> > ### Comment · Reviewer_xbdR · 2025-08-07
> > **Reply to the rebuttal**
> >
> > The authors solved most of my questions and further clarify their framework contirbution to me. This work is still more like an engineering work to me. I decided to keep my score as a Borderline accept.

---

### Author Response · Authors · 2025-08-05
**Overall Response to Reviewers**

We sincerely appreciate all reviewers for their constructive feedback and positive recognition of our work. We are particularly grateful for acknowledgments that our *OmniMotion dataset is valuable* (Reviewer xbdR, zsJy) and *fills a practical gap* (Reviewer PaWz, bxuM). We also appreciate the recognition of our *extensive experiments and ablation results* (Reviewer xbdR, PaWz, zsJy), the *thoughtful consideration behind our LLM-based prompt rewriting feature* (Reviewer PaWz, zsJy), and that the paper is *well-structured and easy to follow* (Reviewer PaWz, bxuM).

We encourage the reviewers to **holistically** evaluate our significant contributions to the field, including:

* **New Form of Text Input**: We introduce the *first* investigation into the critical, yet underexplored, role of expressive text input for text-to-motion generation. This includes annotating motions with enriched details and leveraging LLMs to rephrase user prompts for enhanced precision and expressiveness.

* **New Dataset**: We release OmniMotion, a large-scale, high-quality 3D human motion dataset featuring expressive annotations and maintaining temporal continuity. Our dataset is directly convertible to BVH files, addressing several practical gaps in motion synthesis.

* **Comprehensive Benchmark**: We meticulously developed evaluation metrics and conducted extensive comparative analyses across diverse baselines and ablations. We further enhanced our evaluation model with more compact motion features.

* **Improved Method**: We introduce an improved motion quantization method that maintains high fidelity while being significantly more efficient than previous state-of-the-art methods, requiring less than half the tokens. Coupled with a unified generative masked model, our MoMask++ offers greater flexibility in selecting important tokens during iterative generation, leading to notable performance gains when handling **complex and long text prompts**.

During the rebuttal period, we conducted additional experiments and analyses to address reviewer comments:

1. We investigated how tokens at different scales are prioritized during MoMask++ generation (detailed in A3 to Reviewer PaWz).
2. We provided a performance comparison for different positional encoding strategies (detailed in A4 to Reviewer PaWz).
3. We evaluated the performance gain of LLM-based text augmentation (detailed in A1 to Reviewer zsJy).

We thank all reviewers for their efforts in improving the quality and clarity of our work. As the discussion phase concludes, please do not hesitate to let us know of any additional comments.

---

### Decision · Program_Chairs · 2025-09-17

**Decision:**

Accept (poster)

**Comment:**

This paper was reviewed by four experts in the field, with 4 Borderline Accept recommendations. All the reviewers appreciate that this paper proposes an important dataset for human motion generation, with reasonable technical improvements and relatively comprehensive experiments. Thus, this paper achieves the NeurIPS acceptance requirement. AC highly urges the authors to go through the detailed comments carefully to polish the writing and provide extra experimental details, so as to ensure the final acceptance.